# Denoising of Nifti (MRI) Images with a Regularized Neighborhood Pixel Similarity Wavelet Algorithm

**DOI:** 10.3390/s23187780

**Published:** 2023-09-10

**Authors:** Romoke Grace Akindele, Ming Yu, Paul Shekonya Kanda, Eunice Oluwabunmi Owoola, Ifeoluwapo Aribilola

**Affiliations:** 1School of Electronics and Information Engineering, Hebei University of Technology, Tianjin 300401, China; 201641401001@stu.hebut.edu.cn (P.S.K.); 201940000018@stu.hebut.edu.cn (E.O.O.); 2School of Artificial Intelligence, Hebei University of Technology, Tianjin 300401, China; 3Software Research Institute, Technological University of the Shannon, Midlands Midwest, Co. Westmeath, N37 HD68 Athlone, Ireland; i.aribilola@research.ait.ie

**Keywords:** magnetic resonance imaging (MRI), Gaussian noise, Rician noise, regularized pixel 16 detection, wavelet transform, denoising

## Abstract

The recovery of semantics from corrupted images is a significant challenge in image processing. Noise can obscure features, interfere with accurate analysis, and bias results. To address this issue, the Regularized Neighborhood Pixel Similarity Wavelet algorithm (PixSimWave) was developed for denoising Nifti (magnetic resonance imaging (MRI)). The PixSimWave algorithm uses regularized pixel similarity detection to improve the accuracy of noise reduction by creating patches to analyze the intensity of pixels and locate matching pixels, as well as adaptive neighborhood filtering to estimate noisy pixel values by allocating each pixel a weight based on its similarity. The wavelet transform breaks down the image into scales and orientations, allowing a sparse image representation to allocate a soft threshold on its similarity to the original pixels. The proposed method was evaluated on simulated and raw T1w MRIs, outperforming other methods in terms of an SSIM value of 0.9908 for a low Rician noise level of 3% and 0.9881 for a high noise level of 17%. The addition of Gaussian noise improved PSNR and SSIM, with the results indicating that the proposed method outperformed other models while preserving edges and textures. In summary, the PixSimWave algorithm is a viable noise-elimination approach that employs both sparse wavelet coefficients and regularized similarity with decreased computation time, improving the accuracy of noise reduction in images.

## 1. Introduction

Typically, a medical image dataset includes either a single or multiple images that depict the projection of an anatomical volume onto an image plane (projection or planar imaging), a series of thin slices via a volume (tomographic or multislice two-dimensional imaging), a collection of data from a volume (volume or three-dimensional imaging), or a dynamic series of acquisitions produced by capturing multiple tomographic or volume images over time (four-dimensional imaging). Currently, medical image formats include Analyze, Neuroimaging Informatics Technology Initiative (Nifti), Montreal Neurological Institute (Minc), and Digital Imaging and Communications in Medicine (Dicom), in which the file details show the organization of the image data within the file and instruct the implemented software on how to properly load and display the pixel data [1]. As an improvement from the Analyze format, the Nifti format was developed to resolve the shortcomings in medical imaging data storage. Although initially designed for neuro-imaging purposes, Nifti can also be applied in other domains. One of Nifti’s key characteristics is the inclusion of raw data within the 3D image, which comprises two affine coordinates that connect the voxel index with the spatial index. Nifti offers a benefit over managing multiple Analyze files by arranging two files per 3D scan [2].

The image format is the method used to store data inside images. Additional images are included in the medical image dataset to depict the three-dimensional view. The number of pixels in the horizontal and vertical directions, the number of bits per pixel, and the number of images per subject are all indicated in medical images. Additionally, translating medical formats into digital formats is necessary to process and view the images. Researchers must convert medical imaging formats, which can be accomplished using open-source software tools like Mango—short for Multi-image Analysis GUI (University of Texas Health Science Center at San Antonio, Texas), MRIcron (v1.0.20190902 at the McCausland Center for Brain Imaging, University of South Carolina), 3D slicer (An Image Computing Platform for the Quantitative Imaging Network, Earth, Texas, United States), MicroDicom (founded by Simeon Antonov Stoykov based in Sofia, Bulgaria.), Freesurfer (Athinoula A. Martinos Center for Biomedical Imaging, Charlestown, MA 02129, United States), etc. [2].

Magnetic resonance imaging (MRI) is one sort of image acquired in medical imaging. It is utilized to view the body’s interior structures, such as the brain. These images may assist doctors in identifying abnormalities in a patient, such as a brain tumor, or in assessing the effectiveness of brain surgery. However, noise and other types of deterioration like blurring effects are frequently introduced into MRI during acquisition [3]. The images acquired must be preprocessed before harnessing them for further post-processing like segmentation and classification. Thus, image denoising aims to remove noise with its associated degradations from a corrupted image, thereby restoring the true image [4]. Noise includes several other high-frequency attributes like edges and texture. It is a challenging task to differentiate within the method of denoising and, therefore, the denoised images may inevitably lose some details [5]. The denoising of images is a key topic in the arena of image processing, specifically medical imaging, as it provides a means to reduce the noise present in an image, improve its quality, and make it more suitable for additional analysis. Denoising is becoming increasingly popular as it helps to improve the robustness of medical image analysis. This is mandatory because, in the process of acquisition of the images, there may be the presence of artifacts, noise, blurriness, etc., which may inhibit the accuracy of the diagnosis of the disease. The principal challenge comprises lessening the amount of noise, that is, regularizing the MRI while protecting the subtleties, the edges, and the little structures that could be pivotal for a correct diagnosis. Nevertheless, given that local denoising techniques are simple in terms of low time complexity, their capabilities are limited when the noise level is large. Antecedent to this is the fact that a high noise level substantially disrupts the correlations of the surrounding pixels. Non-local self-similarity (NSS) priors have lately been used in several approaches [6] since images contain many comparable patches in different places. Denoising images aim to lessen the noise in the acquired medical images while attenuating the loss of inherent attributes and amplifying the signal/noise (SNR) [7]. Moreover, the NLM filter suggested by [8] has received interest in denoising MRI due to the abundance of repeated patterns in both naturalistic and medical imaging. The majority of the state-of-the-art methods yielded good results; however, the computation time is increased, and the original attributes of the images are lost in the process.

Considering the facts stated above, our paper proposes the PixSimWave algorithm to tackle these challenges. Utilizing this approach reduces the computation time by exploiting the regularized neighborhood pixel similarity alongside the wavelet coefficient sparsity. As such, it divides the image into patches and compares the pixels’ intensity to find similar pixels. Then, a weight is assigned to each pixel based on the similarity to the original pixels. The wavelet transform decomposition breaks down the image into different scales and orientations, resulting in a sparse representation of the image. This means that only a small percentage of the coefficients are non-zero, while the rest are close to zero. Our algorithm falls under the category of filtering and regularization techniques using neighborhood pixels and not diffusion–denoising generative models.

The contributions of this research are as follows:A low computational complexity was achieved with the PixSimWave algorithm. By dividing the images into patches, different levels of detail can be captured, thus reducing the number of pixel comparisons required. Hence, reducing complexity by narrowing down the size of the data that the algorithm operates on results in faster and more efficient denoising of images.The edges and other high-frequency features of the images were preserved when the PixSimWave algorithm was applied for denoising, as it offers the ability to both reduce noise and preserve features simultaneously. The reason for this is that it can isolate and distinguish noise components while safeguarding the important features of the images.Evaluation of the PixSimWave algorithm demonstrates a high PSNR when compared with other algorithms, which proves its efficiency for denoising medical images.The PixSimWave algorithm can be applied to images of any size and resolution, making it a versatile method for image denoising.

In the rest of this paper, we present the following information. Section 2 explores the current research status and current trends in image conversion, as well as the various noises that can be applied and the different tools available for removing them. Section 3 discusses the algorithm’s process flow, implementation steps, pseudocode, and its computational complexity. Throughout Section 4, the experiment with results of the research and a discussion of the findings can be found, while a summary of the work is provided in Section 5.

## 2. Literature Review

This section analyzes the existing literature and research gaps on image conversion, different noises that can be applied, and different tools that can be used to remove these noises.

### 2.1. Variation of Noise

In image processing, noise refers to random variations in pixel values that are not representative of the true content of the image. It is often caused by external factors such as electromagnetic interference, sensor noise, or errors in image acquisition. Noise can significantly degrade the quality of an image by obscuring important information, reducing contrast, and introducing artifacts. Several types of noise can occur in digital images, but the most important ones will be discussed in this paper.

#### 2.1.1. Gaussian Noise

Gaussian noise is the most prevalent type of noise in digital imaging and arises due to sensor constraints in the course of collecting the images under low luminance intensity circumstances, resulting in a challenging task for the visible light sensors to record scene information correctly [9].

#### 2.1.2. Rician Noise

Rician noise occurs when the signal amplitude is superimposed with Gaussian noise that has the same standard deviation as the signal. This type of noise is common in magnetic resonance imaging (MRI) and synthetic aperture radar (SAR) images. It can make images look grainy and reduce their quality. Therefore, it is important to remove noise from the images or minimize it during the acquisition process.

#### 2.1.3. Impulse Noise (Salt and Pepper Noise)

This noise statistically drops the original value of the data. In addition to salt and pepper noise, this type of noise is sometimes referred to as white noise. Nevertheless, some pixel values in the image are altered by salt and pepper noise rather than the entire image being corrupted. There is a possibility that some neighbors will remain the same in a noisy image.

#### 2.1.4. Speckle Noise

This noise is multiplicative in nature. These are visible in coherent imaging systems such as lasers, radars, acoustics, etc. The occurrence of speckle noise in an image is similar to that of Gaussian noise [10].

#### 2.1.5. Poisson Noise

This is a type of noise that is often encountered in digital images. It is caused by the random variation in the number of photons detected by a camera sensor or other imaging system. Poisson noise tends to be more prevalent in low light conditions, where the number of photons is reduced and the noise is more visible. In image processing, the Poisson distribution is often used to model the statistical properties of the noise [10].

However, in MRI, as our case study, the existing literature [11] has established that the prominent noises present in it are Gaussian and Rician noise. There are random variations during the acquisition process of MRI due to the short acquisition time, as well as interference caused by internal components of the MR scanner. As a result of these variations, we can model them as Gaussian and Rician noises, respectively. Noise with a Gaussian distribution has the same probability distribution as additive noise. Alternatively, Rician noise is non-additive and tends to produce Rician-distributed image data. As signal-to-noise ratios (SNRs) increase, Rician distributions tend to resemble Gaussian distributions [12,13,14,15].

### 2.2. Denoising Algorithms

The assessment of noise and image denoising in MRI has been a significant field of research for some years, employing numerous techniques. These can be roughly divided into four categories: filtering methods, transform domain methods, statistical modeling methods, and regularization methods.

Filtering methods: In eliminating contrastive noise constituents, multifarious filtering algorithms have been propounded. The exploration of MRI denoising approaches was demonstrated by the usage of smoothing using a Gaussian filter in voxel-based morphometry (VBM) analysis. This was used as a preprocessing step before partitioning the grey matter of the MRI for discrepancy artifacts [16]. A Wiener filter was utilized, employing familiarized orientation to deduce the structure in every voxel and generating reformed parameters by adaptively merging the techniques iteratively. This was locally experimented on an MRI brain phantom, assisting in the segmentation algorithm to extract more exquisite details [17]. In Ref. [18], a dynamically weighted adaptive median filter (ADWMF) was proposed as an impulsive noise removal filter. An ADWMF filter is weighted dynamically based on the results of noise detection, instead of fixed weights. Both low and high-density images perform better when using the AMWMF algorithm. By combining geometric, photometric, and local structure similarities, a trilateral filter was proposed in [19], yielding edge-preserving results. The algorithms proposed by [20] were based on the idea of incorporating as many structural similarities as possible. Notwithstanding, the method was time-consuming and insufficient in terms of searching pixels, while Ref. [21] proposed an innovative approach for MRI denoising that incorporated the non-local means filter, Wiener filter, and median filter. Although it was more accurate than NLM, there was an increment in the computation time.Transform domain methods: Transform domain filtering methods are signal processing techniques that operate on signals in a different domain, often by transforming them into a representation that emphasizes specific features or properties. For instance, Bayesian Markov random field models [18], rough set and kernel PCA [19], and higher-order singular value decomposition [20,21] introduce an integrated framework combining wavelet-based processing and statistical testing in the spatial domain. It proposes two enhancements: revisiting the paradigm and reducing spatial bias, and compensating for wavelet transform shift-invariance. Furthermore, Ref. [22] introduced a technique for locating neuronally related fluctuations in fMRI data which automates noise detection and produces discrete spatial and temporal features for effective cleanup, while Karnati et al. [23] proposed higher-order Partial Differential Equations (PDEs), an image smoothing method using a fourth-order PDE model. In MRI denoising, noise estimation methods in the wavelet domain are also utilized, where MRI is divided into sub-bands at various scales in the wavelet domain. To estimate signal components, coefficients are treated using soft or hard thresholding [22]. With variant correspondence, the region of proximity between two pixels can be computed using multi-area tables and the fast Fourier transform, resulting in a 50-fold speed-up while retaining comparable quality [23]. There are several methods for wavelet-based image denoising, including Bayesian shrinkage, soft thresholding, and wavelet packet thresholding, each with varying approaches to estimating the threshold and reassembling the image [24,25]. In contrast to traditional methods like Fourier-based filtering, wavelet-based denoising is more adept at retaining image details and high-frequency properties due to the sparsity of wavelet coefficients in which most information is concentrated in just a few coefficients [26].Statistical modeling methods: In Ref. [27], first-order statistics were proposed for the frequency content of the median filtered residuals (MFRs) of original and median filtered images for use in image forensics. The resulting feature set is significantly larger than the deep learning-based detector and delivers better detection results in low-resolution images of all quality levels. Conversely, Kazerouni et al.’s [28] diffusion models are probabilistic generative models that learn complex distributions by adding noise to the data and then restoring the original structure. This allows for accurate modeling of data distributions affected by random noise. Chung et al. [29] proposed a new denoising method using score-based reverse diffusion sampling that overcomes drawbacks and excels in in vivo liver MRI data with complex noise mixtures. Wu et al. [30] developed a deep-learning framework for super-resolution brain MRI images that included self-attention. The results of the experiments showed that based on the learned perceptual image patch similarity (LPIPS) metric, their framework produced the least distorted super-resolution brain MRI images.Regularization methods: Regularization is a popular technique used in denoising algorithms to reduce noise while preserving important details in images or signals. As an illustration, Rudin et al. [31] introduced a nonlinear total variation-based noise removal algorithm by using Lagrange multipliers and the gradient-projection approach, a limited optimization numerical algorithm that reduces image noise, and a non-invasive method that produces cutting-edge outcomes. Also, Manjón et al. [32] proposed an adaptive non-local means of noise removal by spatially sifting out the intrinsic noise in the MRI. By regulating the filtering parameter, they were able to identify MRI data with an exact pattern via contrasted mean levels, rectifying its intensity in-homogeneity while attenuating the error in the noise variance of which several methods for speeding up execution were presented. It is possible to compute the mean for each pixel by searching just the pixel itself rather than the entire image. Likewise, deep learning methods have achieved state-of-the-art performance in denoising tasks. One notable approach [33] exploits the structure of the neural network itself to perform denoising, without the need for large training datasets. In addition, GANs have shown promising results in denoising tasks by training a generator network to transform noisy images into clean ones. Among others is the Noise2Noise (N2N) method introduced by [34]. This method trains a network using only pairs of noisy images, without the need for clean reference images.

Fully aware that the impact of noise appended can be greatly weeded out without any iota of recognition depending on the efficiency of filtration and denoising algorithms exploited, our proposed algorithm aims to use less computation time by exploiting the sparsity of the wavelet coefficients. This involves quantifying the gray values of each pixel by selecting the features that eliminate noise or degradation in the image based on their self-similarity regularization and then convolving the image with a series of wavelet functions at different scales and positions. Decomposition of the image into multiple frequency components using a selective wavelet function enhances simultaneous noise reduction and feature preservation, thus making it a versatile method that can be applied to images of any size and resolution.

## 3. PixSimWave Methodology

This section discussed the process flow, implementation steps, pseudocode, and the computational complexity of the PixSimWave algorithm.

### 3.1. Process Flow of PixSimWave Algorithm

The process flow of the PixSimWave algorithm is illustrated in Figure 1. This algorithm loads an image that contains noise from the path and divides this image into two patches for easy comparison. These two patches are compared with each other based on their pixel intensity to determine the similarities between them. After extracting the similarities, a weight is applied to these pixels and hence an average of this weight is computed. The computed average weight was also converted to a wavelet domain and a soft thresholding was applied, as well as the inverse wavelet transform to remove the noise from the image. Using this method, the final result was a noiseless image that preserved the image’s features and required less computing time to process.

#### 3.1.1. Regularized Pixel Similarity Detection

Regularized pixel similarity detection was used for adaptive neighborhood filtering to improve the accuracy of noise reduction in the image. This technique estimated the value of a noisy pixel by identifying synonymities between different pixel regions within the image.

Additional constraints are added to the similarity measurement process to improve the accuracy of the estimate and reduce the impact of outliers and noise in the image. This approach involves using a weighted average of pixel values within a given area, but with additional smoothing and regularization factors that ensure the resulting estimate is more accurate and reliable.

By performing a regularized pixel similarity detection process, better results, even when working with images that contain high levels of noise, are achieved.

#### 3.1.2. Application of Wavelet Transform

Utilizing a time–frequency representation and thresholding method, wavelet transform is an effective technique for reducing noise in images while maintaining essential features. During the image denoising process, wavelet transform is used to decompose the image into coefficients at varying scales and orientations, which are then thresholded and reassembled to create a cleaner image.

### 3.2. Implementation Steps of PixSimWave

Step 1. Load the image frame:

The image is loaded from its path location in the system and read the pixels on the height and width as shown in Equation (1) where *x* is the height and *y* is the width.
(1)x,y ⇐ image

Step 2. Pixel extraction:

The image is divided into patches *a* and *b* to extract similarities. The pixel value of patch *P*(*a*) at position *x*, *y* is shown in Equation (2) while patch *P*(*b*) at position *x*, *y* is shown in Equation (3), respectively.
(2)P(a) ⇐ image(x,y)
(3)P(b) ⇐ image(x,y)

Step 3. Pixel comparison:

In this step, the pixel intensity values of Equations (2) and (3) as shown in Equation (4) are compared to find similar patches.
(4)$$P(compare)\;⇐\;P_{a}\left(x,y\right)-P_{b}\left(x,y\right)$$

Step 4. Weight calculation:

A weight is assigned to the acquired similar patches or pixels from Equation (4), which results in Equation (5). The L2 norm was chosen because of the closeness between the two vectors, and it measures the difference between the original image and the denoised image.
(5)$$W(a,b)=\frac{1}{\alpha \left(x\right)}\exp\left(-\frac{||P_{a}\left(x,y\right)-P_{b}\left(x,y\right)|{|}_{2,\eta }^{2}}{h^{2}}\right)$$
where W(*a*,*b*) is the indices of patches in the image, α(x) is a normalization constant, *η* is the standard deviation, *P*(*a* + *x*, *y*), *P*(*b* + *x*, *y*) are the pixel values of patches *a* and *b* at position (*x*,*y*), and *h* is a parameter controlling the degree of smoothing. We experimented with the choice of parameters in terms of smoothing scale, search window, and kernel sizes to be used. After this, the optimized values were derived via preferential experiments. As a result, the smoothing scale, search window, and kernel sizes were set as 3 × 3, 7 × 7, and 21 × 21, respectively for all the levels of noise added.

Step 5. Weighted average:

Perform an average on the calculated weight in Equation (5), as described in Equation (6), which gives the weighted average of the extracted patch values.
(6)$$\hat{f}\left(x,\;y\right)=\frac{1}{Z\left(y\right)}\;\sum x\;\in \;Ω\left(y\right)W\left(a,\;b\right)\times \hat{g}\left(x,\;y\right)$$
where $\hat{f}$(*x*, *y*) is the denoised pixel value at position (*x*, *y*), and *Z*(*y*) is the normalization factor, which is the sum of weights given to each pixel in the patch Ω(*y*). Ω(*y*) is the patch centered around pixel *y*. *Z*(*y*) is the normalization factor, which is the sum of weights given to each pixel in the patch Ω(*y*). $\hat{g}$(*x*, *y*) is the original pixel value at position *x*, *y*. W(*a*, *b*) is the weight computed between pixels *x* and *y*, which is a function of the distance between their patch vectors and the value of a smoothing parameter *h*.

Step 6. Convert to wavelet domain:

This further separates the noise from the pixel into different frequency bands by using a chosen wavelet basis function and multiresolution analysis as shown in Equation (7).

The wavelet basis described in Equation (7) was chosen because the coefficients of the resulting signal decomposed, and this was used to reconstruct the image signal. Due to their ability to represent signal features at different scales, they are effective for noise removal. The localized nature of wavelets allows for the separation of signal and noise components, enabling the removal of unwanted noise while preserving important signal features.
(7)$$\psi a,b^{\left(t\right)=\frac{1}{\sqrt{a}}\;\psi \left(\frac{t-b}{a}\right)}\;\left[W_{\psi }f\right]\left(a,b\right)=\frac{1}{\sqrt{ab}}\;\iint \hat{f}\left(x,y\right)\psi \left[\frac{\left(x-a\right)}{b},\;\frac{\left(y-b\right)}{b}\right]\frac{dx}{dy}$$

Step 7. Apply soft thresholding:

Soft thresholding was applied to the coefficients of the wavelet domain in Equation (7), which reduces or removes small amplitude noise while preserving the significant features of the image as described in Equation (8).
(8)$$\hat{W}\left(a,b\right)=\{\begin{array}{c} sgn\left(W_{a,b}\right)\left(\left|W_{a,b}\right|-\tau \right),\;\;\;\;\;\;\;\;\left|W_{a,b}\right|\geq \tau \\ 0,\;\;\;\;\;\;\;\;\;\;\;\;\;\;\;\;\;\;\;\;\;\;\;\;\;\;\;\left|W_{a,b}\right|<\tau \end{array}$$

Step 8. Apply inverse wavelet transform:

Finally, an inverse wavelet transform was applied to the thresholded coefficients in Equation (8) to give a noiseless image as seen in Equation (9).
(9)$$\hat{f}\left(x,y\right)=\sum \sum \hat{W}\left(a,b\right)\eta \left(\left|\hat{W}\left(a,b\right)\right|-\tau \right)\;$$

### 3.3. Computational Complexity of PixSimWave Algorithm

This section describes the pseudocode in Algorithm 1 and the computational complexity of the proposed PixSimWave algorithm. The PixSimWave Algorithm has multiple steps. **Step 1** loads the image for processing and, therefore, consumes *Big*(*O*) = *n* for *n* number of image frames. **Step 2** divides the image into two patches, which takes *Big*(*O*) = *frame_dimension*. **Step 3** computes the similarities using *Big*(*O*) = *n* for *n* number of pixels. **Step 4** applies the weight using *Big*(*O*) = *n* for *n* number of pixels. **Step 5** averages the computed weight on the pixel with *Big*(*O*) = *n* for *n* number of pixels. **Step 6** converts the pixels to a wavelet domain using *Big*(*O*) = *n* for *n* number of pixels. **Step 7** applies soft thresholding by looping through the pixels to divide the pixels based on the threshold thus the time complexity is *Big*(*O*) = *frame_dimension*. **Step 8** applies inverse wavelet on the threshold pixels, which consumes *Big*(*O*) = *n* for *n* number of pixels. This sums up to *Big*(*O*) = 6*n* (2 × *frame_dimension*), which is a linear time complexity.
*Step* 1 (*S*1): *Big*(*O*) = *n // n images*
*Step* 2 (*S*2): *Big*(*O*) = *frame_dimension*
*Step* 3 (*S*3): *Big*(*O*) = *n // n pixels*
*Step* 4 (*S*4): *Big*(*O*) = *n // n pixels*
*Step* 5 (*S*5): *Big*(*O*) = *n // n pixels*
*Step* 6 (*S*6): *Big*(*O*) = *n // n pixels*
*Step* 7 (*S*7): *Big*(*O*) = *frame_dimension*
*Step* 8 (*S*8): *Big*(*O*) = *n // n pixels*
Tsum=[S1+S2+S3+S4+S5+S6+S7+S8]=[n+frame_dimension+n+n+n+n+frame_dimension+n]
Big(O)=6n(2×frame_dimension)

**Algorithm 1:** Pseudo-code of PixSimwaveInput:     A noise MRI imageOutput:  A noiseless MRI imageData:     Load image from pathimg  ←   image from path/* *Step1: Regularised Pixel Similarity Detection */*patchA   ←  img(x, y)patchB   ←  img(x, y)same_pix   ←   patchA–patchB weig  ←  cv.fastNIMeansDenoising (same_pix, None, 3, 7, 21)ave_weig  ←  sum(weig) × img/* *Step2: Application of wavelet transform */*threshold  ←  0.005 × ave_weigsoftness ←  0.0img_without_noise  ←  wavelet(threshold, softness)save_img_to_folder  ←  img_without_noise**Output:**  A noiseless MRI image

## 4. Experiments, Results, and Discussion

This section analyzed the experiment setup to design PixSimWave, the results of PixSimWave when applied to the image dataset with errors, the statistical analysis of the image obtained after the application of PixSimWave, and the comparative analysis of PixSimWave with other denoising algorithms. In order to develop the proposed PixSimWave algorithm, the OpenCV vision library was utilized in Python programming language. In the setup specification in Table 1, the details of the experiments are provided. The experiments were executed on the dataset from [35]. The dataset contains three types of brain image data: T1-weighted MRI, T2-weighted 332 MRI, and proton density (PD) weighted MRI. Each type of image comes in 181 slices of 256 × 256 pixels, with 1 mm resolution in the x, y, and z planes. The images were simulated to closely resemble real MRI scans, with realistic levels of noise and intensity variations, and anatomical structures that match those found in the human brain. A T1-weighted MRI volume was used as the ground truth to study the efficacy of the proposed method. In a similar vein, we validated the efficiency of our proposed method using real medical T1w MRI images by randomly selecting seven subjects from the OASIS cross-sectional dataset [36] and evaluating the denoising metrics stated in the paper.

### 4.1. Visual Result of PixSimWave Algorithm

This section discussed the visual results of the PixSimWave algorithm when noise was applied to the images. Gaussian and Rician noise were applied to the Nifti MRI images.

#### 4.1.1. Addition of Rician Noise to Nifti Images

Rician noise is a type of noise that is often present in medical images, such as MRI scans, caused by the random interference of multiple signal components. In an image, these signal components could be the result of different tissues or organs, or the contrast agent used in the imaging process. The interference between these components creates a random noise pattern that is superimposed on the image signal. Rician noise can be modeled as a Gaussian noise with a zero mean, added to the image signal that has a non-zero variance. Assuming an input image I(x, y) and a Rician noise component N(x, y) with parameters σ and ση, the resulting noisy image is modeled in Equation (10):(10)$$I^{\prime }\left(x,y\right)=\sqrt{{\left(\left(I\left(x,y\right)+\sigma \eta \right)\right)}^{2}+N{\left(x,y\right)}^{2}}-\frac{\sigma }{\eta }$$
where *σ* is the standard deviation of the Gaussian component of the Rician noise (associated with the real and imaginary parts of the signal) and *ση* is the standard deviation of the Rayleigh component of the Rician noise (associated with the magnitude of the signal). Note that this equation assumes that the noise is added to the magnitude image rather than to the real and imaginary parts separately. To add the noise to the real and imaginary parts, one would need to sample two independent Gaussian distributions with standard deviation *σ*/√ 2 and add the resulting values to the real and imaginary parts of the image, 356, respectively. Rician noise of 5% was added to the original image of slice 20, as displayed in Figure 2. It presents the visual representation of the original, noisy, and denoised images used. Our algorithm performs better with distinct clarity of image quality and good resolution. As depicted in Figure 2h, it can be seen that the proposed method surpasses the other state-of-the-art (SOTA) methods with the edges and textures of the images being preserved. Also, there is a kind of resemblance between the original image and the denoised image of our proposed method, showing correlative robust results.

#### 4.1.2. Addition of Gaussian Noise to Nifti Images

It is a type of statistical noise that has a Gaussian distribution (normal). In other words, the noise values follow a normal Gaussian distribution. The original image is distorted by adding Gaussian noise. A normal distribution’s probability density function matches the probability density function of Gaussian noise, which is essentially statistical [37] as displayed in (11). Let *I*(*x*, *y*) be the original image and *N*(*x*, *y*) be the Gaussian noise added to it. Then, the mathematical equation would be as follows:(11)$$I^{\prime }\left(x,y\right)=I\left(x,y\right)+N\left(x,y\right)$$
where *I*′(*x*, *y*) is the resulting image with Gaussian noise. The value of the noise *N*(*x*, *y*) at each pixel (*x*, *y*) is drawn randomly from a normal distribution with mean 0 and standard deviation σ, and added to the corresponding pixel value of the original image *I*(*x*, *y*). This process is repeated for each pixel in the image to add Gaussian noise to the entire image. Gaussian noise of 5% was added to the original image of slice 20, as displayed in Figure 3. This figure presents the visual representation of the original, noisy, and denoised images used. Our algorithm performs well with the image quality and a higher-grade resolution depicted in Figure 3h, illustrating how our proposed method outperforms other state-of-the-art techniques by preserving edges and textures. Additionally, the denoised image from our algorithm shows a strong correlation with the original image, indicating robust results.

### 4.2. Statistical Analysis of PixSimWave Algorithm

This section analyzed the structural similarity index measure (SSIM), peak signal-to-noise ratio (PSNR), root mean square error (RMSE), and the feature similarity index measure (FSIM) of the proposed PixSimWave algorithm.

#### 4.2.1. Structural Similarity Index Measure (SSIM)

The Structural Similarity Index Method (SSIM) is a model based on perception. According to this concept, structural information is perceived differently when an image degrades on account of preprocessing like data compression or transmission losses. Additionally, it works together with some other crucial perception-based facts for luminance masking and contrast masking. Strongly interdependent pixels or spatially confined pixels are highlighted by the phrase structural information [38]. Unlike PSNR, SSIM evaluates the image structure and is a better evaluation metric for image quality because it uses an uncompressed or distortion-free image as the basis for the evaluation. It evaluates the simile between two images: the actual and the predicted [39], which is expressed in (12).
(12)$$\mathrm{SSIM}\left(x,y\right)=\frac{\left(2\mu_{x}\mu_{y}+C_{1}\right)\left(2\sigma_{xy}+C_{2}\right)}{\left(\mu_{x}^{2}+\mu_{y}^{2}+C_{1}\right)\left(\sigma_{x}^{2}+\sigma_{y}^{2}+C_{2}\right)}$$
where x = original image, y = denoised image with PixSimWave algorithm, *µ_x_* = average of x, *µ_y_* = average of y, $\sigma_{x}^{2}=$ the variance of x, $\sigma_{y}^{2}=$ the variance of *y*, $\sigma_{xy}$= the covariance of *x* and *y*,$\;C_{1}={\left(K_{2}L\right)}^{2}$, $C_{2}={\left(K_{2}L\right)}^{2}$, L = dynamic range, $\left(K_{1}\right)$ = 0.01, and $\left(K_{2}\right)$ = 0.03.

As long as the SSIM of the reconstructed image to the ground truth image is close to 1, one can be sure the image is of good quality. The highest SSIM signifies the best denoising technique. SSIM is regarded as the best metric so far because it quantifies the perceptual difference between two similar images. Table 2 below compares the SSIM of our proposed model to that of other SOTA results. As is readily seen by the comparison in Table 2, we have the highest values marked in bold letters for all the noise levels tested as compared to previous works. Considering the case of low noise levels added at 3%, the SSIM obtained was 2–3% higher than the other methods. Similarly, for high noise levels added at 17%, our proposed method gave 0.9881 while the next best was only 0.7455. As the noise level increases, the SSIM values decrease because the image quality deteriorates, due to the introduction of additional random variations in the image that are not related to the actual content. This results in a decrease in the signal-to-noise ratio, which is the ratio of the original image to the amount of noise in the image.

Furthermore, Figure 4 presents the graphical representation of SSIM analysis on OASIS images. It is evident that the noise added introduced random fluctuations in the image components, leading to differences between the original and distorted images. With higher noise levels, the luminance component of the images becomes less similar due to the presence of random brightness variations. This is because the contrast component also becomes affected, as noise can blur edges and reduce the distinction between different regions in the image. The structural component, which captures the similarity in local patterns, is similarly impacted by noise, resulting in reduced SSIM. Thus, at a noise level greater than 5–7%, the increased noise tends to degrade the SSIM score, indicating a decrease in perceived image quality.

#### 4.2.2. Peak Signal-to-Noise Ratio (PSNR)

The PSNR is a metric used to estimate how much distortion-causing noise is present between two images and how well it impacts the quality of the display. The signal’s extremely wide dynamic range makes it necessary to compute the PSNR as the decibel scale’s logarithm term. This infers that the decibel (dB) form is used to compute the ratio between both images. For evaluating PSNR, *MSE* is utilized in the process. Mathematically, PSNR is defined in Equation (13).
(13)$$\mathrm{PSNR}=10.\log_{10}\left(\frac{MAX_{i}}{\sqrt{MSE}}\right)$$

*MSE* represents the Mean-Square Error and *MAX_i_* is the maximum possible intensity value of the image. The number 255 corresponds to pixels represented by eight bits per sample. High PSNR means the *MSE* between the actual image and the predicted image is very low. A properly restored image should have a high PSNR value indicating its high quality, while a low PSNR value will result in a very poor quality of the restored image. Although the PSNR is a widely used tool for evaluating image quality, its deficiencies have long been known conventionally. Specifically, PSNR does not take the structural fidelity and edge integrity of the image into account because low PSNR images might nevertheless be of good quality. At the same time, in Table 3, the addition of Gaussian and Rician noise performs relatively better than the previous work, as shown by the increase in PSNR value. As a case in point, let us look at the situation in which the noise level was 7%, and the PSNR was 7–8 dB higher than the SOTA results. Larger noise levels correspond to lower PSNR values, but PixSimWave still has a PNSR value higher than 40. This demonstrates the superiority of our proposed algorithm.

Similarly, the analysis performed on the OASIS cross-sectional datasets selected randomly, as illustrated in Figure 5, indicates the results obtained from the PSNR versus noise level. From the graph, as the noise level increases, the PSNR value decreases. This is because noise introduces random fluctuations and errors into the signal, resulting in a loss of fidelity and information. At 3–5%, the PSNR value is relatively high, indicating that the noise has little impact on the image quality. However, as the noise level increases, the PSNR value decreases, indicating a higher level of distortion and reduced fidelity of the image. This means that the noisy image deviates more from the original, resulting in a lower PSNR value.

#### 4.2.3. Root Mean Square Error (RMSE)

The Root Mean-Square Error is a widely used technique for measuring errors, particularly the differences between actual and predicted values, as shown in Equation (14). This method assesses the scale of the errors and is an excellent tool for evaluating the accuracy of predictions by different estimators for a particular variable [39].
(14)$$RMSE\left(\hat{\theta }\right)=\sqrt{MSE\left(\hat{\theta }\right)}$$

Figure 6 displays the *RMSE* results, which increase exponentially with an increase in noise level, as expected. This is because the noise added random variations to the images, making it harder to accurately predict the outcome. As a result, the errors in the predictions also increase, leading to higher *RMSE* values.

#### 4.2.4. Feature Similarity Index Measure (FSIM)

To fully understand how the FSIM metric works, there is a need to give a brief introduction of two terms associated with it. These are phase congruency and gradient magnitude.

Phase Congruency:

Coupé et al.’s [43] technique aimed to identify visual features. The fact that phase congruency is unaffected by changes in picture brightness is one of its key qualities. Additionally, it can recognize additional intriguing traits. It emphasizes the image’s characteristics in the frequency domain. Contrast is invariant to phase congruency.

Gradient Magnitude:

A widely traditional area of study in digital image processing is the computation of image gradients. The gradient operators are expressed using convolution masks. To measure the gradients, numerous convolutional masks are employed. Assuming I(*x*) is an image, while *H_x_* and *V_y_* are the horizontal and vertical gradients, respectively; therefore, the gradient magnitude I(*x*) is represented in (15) [44] as follows:(15)$$I\left(x\right)=\sqrt{H_{x}^{2}+V_{y}^{2}}$$

Correspondingly, Figure 7 illustrates the FSIM performance on the OASIS real medical T1w MRI to measure the image quality. Unlike RMSE, an increase in noise level decreases the FSIM value. This is because the noise present in an image affects its clarity, sharpness, and overall visual appeal, which is what FSIM aims to measure. The higher the level of noise, the more distorted and less clear the image becomes, resulting in a lower FSIM value.

### 4.3. Real Clinical MRI

Using a clinical right lower leg MRI, Figure 8 shows the performance of the proposed approach. In addition, a zoomed region is displayed for each case. Even small-scale edges can be detected using the proposed method. It is evident that improved denoising performance has been accompanied by enhanced tissue contrast and this does not diminish the difference in brightness across the images.

Additionally, we evaluated our proposed method on another slice of the T1w brainweb and made a comparison with other state-of-the-art methods, as shown in Figure 9.

From Table 4, it is evident that our proposed method surpasses the performance of the four other state-of-the-art methods in MRI image denoising, as evaluated by SSIM metrics. Our method consistently achieves the highest SSIM scores for the MRI images at all noise levels. Particularly, it outperforms the other methods significantly when the noise level in the MRI images exceeds 3–5%.

Furthermore, we experimented with our proposed algorithm on a clinical T1-W cortical MRI joint to joint (MRI LOWER EXTREMITY W/WO CONT) of size 256 × 256 pixels, TE = 8 ms, TR = 466.664 ms, and slice thickness = 5.0 mm; data generated by the TCGA Research Network: https://www.cancer.gov/tcga (accessed on 22 July 2023). Imaging was performed on a Phillips Gemini Medical System.

### 4.4. Computational Analysis of PixSimWave with Other Denoising Techniques

The proposed PixSimWave algorithm was compared with other techniques, as discussed in Table 5.

Table 5 presents the computational times on a T1-W phantom image of 181 × 217 × 181 voxels with 9% noise. In terms of PSNR and the quality of denoised images, the proposed method is better than other approaches.

## 5. Conclusions and Future Work

This paper presented a Regularized Neighborhood Pixel Similarity Wavelet Algorithm (PixSimWave) for denoising Nifti (MRI) Images. By analyzing pixel intensity and dividing the image into patches, the algorithm improves noise reduction accuracy by identifying synonyms and reducing computation time. A performance evaluation of our proposed method was conducted on both simulated and real MRI datasets in comparison with some related state-of-the-art (SOTA) approaches. It was observed that our proposed method achieves high performance, with PSNR and SSIM increasing with the noise level; the features are also preserved, making it a versatile method for image denoising. In addition to this, from the displayed visual representation, the proposed PixSimWave algorithm shows a better improvement with the edges and textures being preserved after eliminating noise alongside other degradations on the MRI.

However, in terms of running time, this method’s computational cost is less inferior to LMMSE. In the future, we will try to decrease computational complexity. We will also investigate strategies to optimize the algorithm further. Additionally, we plan to evaluate the method on other image and noise datasets.

## Figures and Tables

**Figure 1 sensors-23-07780-f001:**
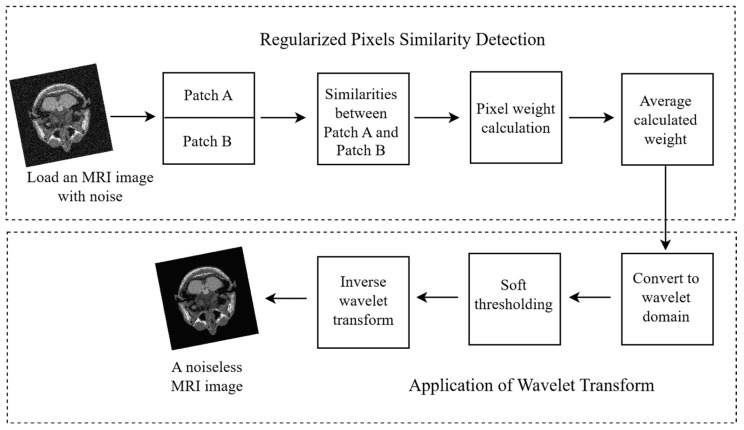
PixSimWave algorithm process flow.

**Figure 2 sensors-23-07780-f002:**
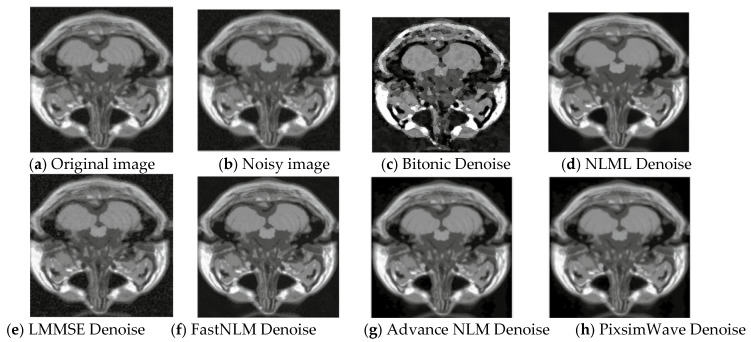
Visual representation of T1 MRI image (Slice 20) with 5% Rician noise. (**a**) Original T1 MRI image, (**b**) 5% rician noisy image, (**c**) Denoised with Bitonic, (**d**) Denoised with NLML, (**e**) Denoised with LMMSE, (**f**) Denoised with FastNLM, (**g**) Denoised with Advance NLM, and (**h**) Denoised with PixSimWave.

**Figure 3 sensors-23-07780-f003:**
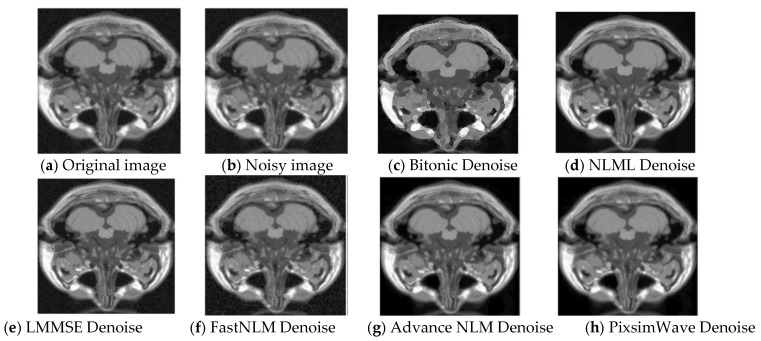
Visual representation of T1 MRI image (Slice 20) with 5% Gaussian noise. (**a**) Original T1 MRI image, (**b**) 5% Gaussian noisy image, (**c**) Denoised with Bitonic, (**d**) Denoised with NLML, (**e**) Denoised with LMMSE, (**f**) Denoised with FastNLM, (**g**) Denoised with Advance NLM, and (**h**) Denoised with *PixSimWave*.

**Figure 4 sensors-23-07780-f004:**
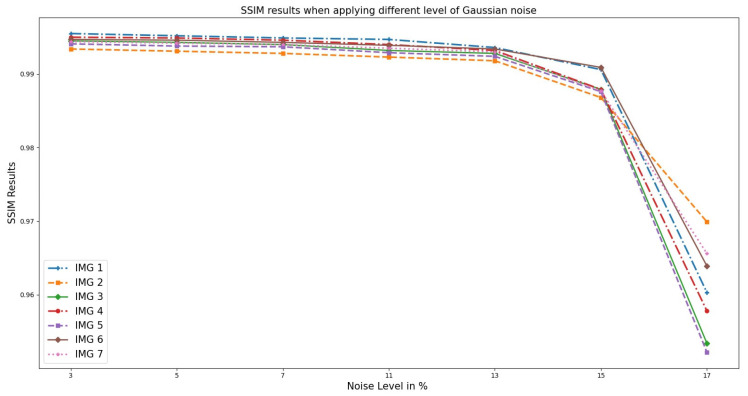
SSIM; the result when different Gaussian noise levels were applied to OASIS images and PixSim Wave was used for denoising.

**Figure 5 sensors-23-07780-f005:**
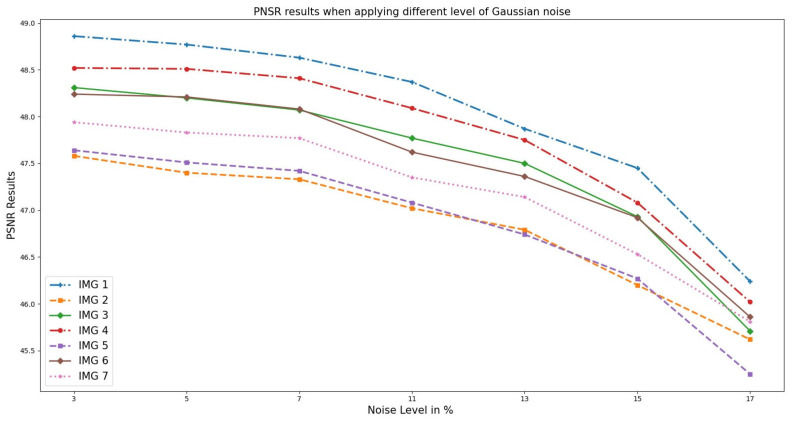
PSNR results when different Gaussian noise levels were applied to OASIS images and PixSimWave was used for denoising.

**Figure 6 sensors-23-07780-f006:**
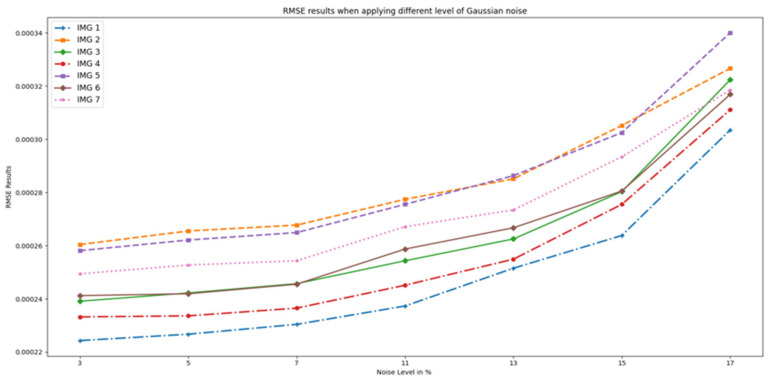
RMSE results when different Gaussian noise levels were applied to OASIS images and PixSimWave was used for denoising.

**Figure 7 sensors-23-07780-f007:**
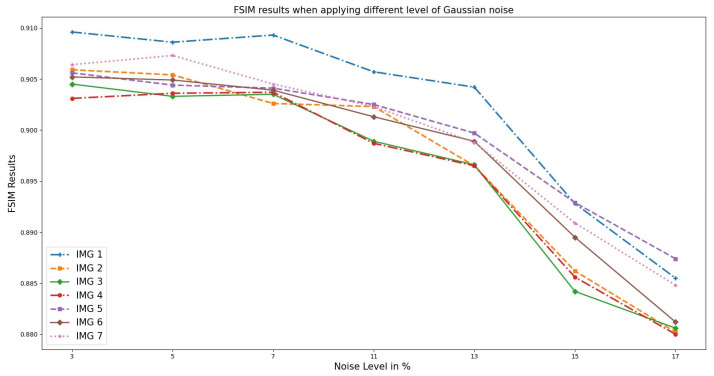
FSIM results when different Gaussian noise levels were applied to OASIS images and PixSimWave was used for denoising.

**Figure 8 sensors-23-07780-f008:**
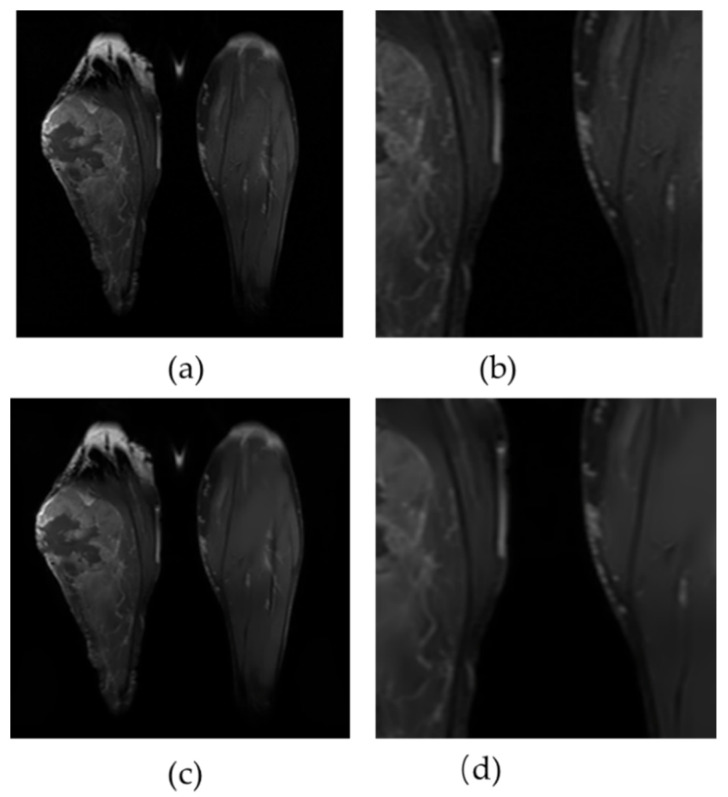
An analysis of the denoising results on Clinical Leg MRI. Figure (**a**,**b**) illustrate the original and zoomed images, respectively. Figure (**c**,**d**) present the correlated results of the proposed method.

**Figure 9 sensors-23-07780-f009:**
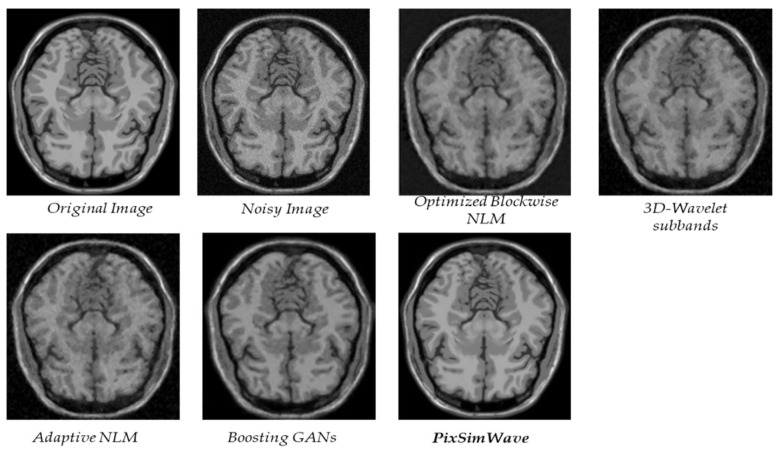
An analysis of the T1w Brainweb datasets to determine the quality of different methods.

**Table 1 sensors-23-07780-t001:** Experimental setup for PixSimWave algorithm.

System/Device	Specification
Processor	Intel(R) Core(TM) i7-10750H CPU @ 2.60 GHz 2.59 GHz
Installed RAM	32 GB
System type	64-bit operating system, x64-based
Operating Systems	Windows 10
Graphics	Intel^®^ UHD graphics

**Table 2 sensors-23-07780-t002:** SSIM Comparison indexes for PixSimWave and other algorithms on Slice 20 using Rician and Gaussian noise with various noise densities.

			Noise Density (%)					
Noise	Techniques	3	5	7	11	13	15	17
Rician	Noisy image	0.8970	0.8804	0.8350	0.7629	0.7427	0.7182	0.6784
	Bitonic [11]	0.9441	0.9345	0.9120	0.8158	0.7785	0.7717	0.7256
	NLML [40]	0.9554	0.9415	0.9200	0.8164	0.7786	0.7726	0.7385
	LMMSE [12]	0.9231	0.9187	0.8870	0.7965	0.7215	0.7528	0.7241
	FastNLM [41]	0.9351	0.9378	0.9137	0.8029	0.7651	0.7635	0.7280
	ANLM [42]	0.9641	0.9561	0.9380	0.8306	0.7900	0.7824	0.7455
	PixSimWave	0.9908	0.9899	0.9895	0.9886	0.9883	0.9882	0.9881
Gaussian	Noisy image	0.9497	0.8854	0.8381	0.7766	0.7507	0.5893	0.5429
	Bitonic [11]	0.9576	0.9248	0.8750	0.8029	0.8140	0.7824	0.7216
	NLML [40]	0.9682	0.9345	0.8850	0.8182	0.7851	0.7890	0.7448
	LMMSE [12]	0.9540	0.9002	0.8530	0.7611	0.7407	0.7299	0.6822
	FastNLM [41]	0.9640	0.9283	0.8680	0.7453	0.7611	0.7961	0.7520
	ANLM [42]	0.9720	0.9453	0.9140	0.8400	0.8226	0.8133	0.7612
	PixSimWave	0.9913	0.9907	0.9895	0.9837	0.9787	0.9727	0.9655

**Table 3 sensors-23-07780-t003:** PSNR Comparison indexes for PixSimWave and other algorithms on slice 20 using Rician and Gaussian noise with various noise densities.

				Noise Density (%)				
Noise	Techniques	3	5	7	11	13	15	17
Rician	Noisy image	37.58	33.21	31.25	28.85	26.13	25.86	24.75
	Bitonic [11]	38.41	35.58	33.25	31.18	29.44	28.92	27.58
	NLML [40]	40.51	36.06	34.52	31.89	30.15	29.18	28.07
	LMMSE [12]	36.67	31.80	30.73	28.51	27.48	26.50	25.59
	FastNLM [41]	38.33	35.82	33.51	30.51	29.08	28.32	27.19
	ANLM [42]	41.04	37.17	35.04	32.33	30.80	30.07	29.31
	PixSimWave	46.80	46.62	46.50	46.35	46.30	46.25	46.24
Gaussian	Noisy image	39.54	37.23	34.48	31.53	30.83	30.20	29.81
	Bitonic [11]	40.85	38.25	35.32	34.88	32.24	32.51	31.62
	NLML [40]	41.51	39.28	38.83	34.76	34.08	33.69	32.89
	LMMSE [12]	40.72	37.66	36.02	32.91	31.92	31.07	30.38
	FastNLM [41]	40.04	38.82	37.19	33.41	33.69	32.73	31.55
	ANLM [42]	42.64	40.47	38.62	36.98	35.45	34.36	33.44
	PixSimWave	46.80	46.59	46.09	44.29	43.02	42.07	41.00

**Table 4 sensors-23-07780-t004:** SSIM Comparison indexes for PixSimWave and other algorithms on another slice with Rician noise and varying noise densities.

		Noise Density (%)				
Noise	Techniques	1	3	5	7	9
Rician	Optimized Blockwise NLM [45]	0.9826	0.9273	0.8710	0.8185	0.7706
	3D-Wavelet subbands [46]	0.9807	0.9236	0.8677	0.8168	0.7715
	Adaptive NLM [32]	0.9747	0.8883	0.8005	0.7190	0.6464
	Boosting GANs [44]	0.9810	0.9763	0.9691	0.9615	0.9540
	PixSimWave	0.9903	0.9835	0.9818	0.9807	0.9802

**Table 5 sensors-23-07780-t005:** Comparison of different techniques in terms of computational time and denoising quality as obtained on a T1-w phantom image of 181 × 217 × 181 voxels with 9% noise.

Techniques	Computational Time (s)	PSNR (dB)
Bitonic	-	-
NLML	-	-
LMMSE	4.92	26.17
FASTNLM	3162	34.19
ANLM	-	-
Optimized Blockwise NLM	135	33.75
3D-Wavelet subbands	181	34.47
Adaptive NLM	-	-
Boosting GANs	-	-
PixSimWave	5.59	44.67

## Data Availability

Data Availability The BrainWeb and OASIS datasets, which are publicly available at BrainWeb: https://brainweb.bic.mni.mcgill.ca/brainweb/ (accessed on 5 June 2023); and OASIS: https://www.oasis-brains.org/ (accessed on 2 May 2023). The TCGA SARC resource website is available at https://tcga-data.nci.nih.gov/docs/publications/sarc_2017/ (accessed on 22 July 2023).

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
