# Peer review of "Denoising of Nifti (MRI) Images with a Regularized Neighborhood Pixel Similarity Wavelet Algorithm"

_sensors, 2023, doi:10.3390/s23187780_

Round 1
Reviewer 1 Report
Overall, the paper titled "Denoising of Nifti (MRI) Images with a Regularized Neighborhood Pixel Similarity Wavelet Algorithm" presents an interesting approach for denoising MRI images. The methodology is well-described, and the evaluation results demonstrate the algorithm's superior performance over existing methods. Here are my specific comments on the paper:
1 The 'where' after the formula should be lowercase and not indented.
2 It is suggested to change P(a+x,y) to Pa(x,y) to reduce confusion for readers.
3 Why does Formula 5 choose L2 norm? And please pay attention to the correctness of the superscripts and subscripts.
4 Different wavelet basis functions are suitable for different types of signals and noise. Why is the wavelet basis described in Formula 7 chosen?
5 You only described the computational complexity of the proposed algorithm. To emphasize the advantages of the proposed algorithm in terms of computational time, please compare the computational time with other algorithms in a table format.
6 Recently, diffusion denoising generative model has achieved good performance in Magnetic resonance imaging (MRI). However, the relevant content was not discussed in the manuscript. Please discuss literatures related to diffusion denoising generative model and add them to Table 2 and Table 3.
7 Tables 2 and 3 only show that the results of Slice 20 are weakly persuasive. More samples and even slices of the entire dataset should be denoised and presented with the average and standard deviation of the metrics.
8 It is obviously not enough to select only three subjects from the OASIS Cross-sectional data set to validate the effectiveness of the proposed algorithm. Please evaluate more cases.
9 With the development of MRI technology, the emergence of multi coil imaging mode makes noise become complex non central Chi-squared distribution noise rather than Rician noise in single coil imaging. Please consider more types of noise in the manuscript.
10 The denoising methods for magnetic resonance images can be roughly divided into four categories: filtering methods, transform domain methods, statistical modeling methods, and regularization methods. Therefore, please cluster the algorithms mentioned in section 2.3.
11 The description of image format conversion appears in the manuscript, which is considered redundant. Your work focuses on denoising, so please reduce the text related to image conversion.
None
Author Response
Response to Reviewer 1 Comments
Point 1: The 'where' after the formula should be lowercase and not indented.
Response 1: Thanks for your observations and contributions. As suggested by the reviewer, we have been able to make the changes where these appear in the manuscript. After Formula 6, the ‘where’ has been changed to lowercase, and the indentation was removed in line 322 in the manuscript.
The same changes have been made to Section 4.1.1, Formula 10, line 411 and Section 4.2.1 of Formula 12 line 463 in the revised manuscript.
Point 2: It is suggested to change P(a+x,y) to Pa(x,y) to reduce confusion for readers.
Response 2: Thank you so much for catching these glaring and confusing errors, which we have now corrected. Accordingly, in the manuscript, we have revised it by switching the order in lines 314, 319 and as requested by the reviewer.
Point 3: Why does Formula 5 choose L2 norm? And please pay attention to the correctness of the superscripts and subscripts.
Response 3:Thanks for your insightful comment and careful inspection of the
manuscript, and we are sorry for the typo error as regards the superscripts and subscripts. We choose the L2 norm because it is a common measure of the "closeness" between two vectors. In the context of our proposed denoising algorithm, the L2 norm is used to measure the difference between the original image and the denoised image. Also, in our denoising method, we assume that the noise follows a Gaussian distribution, and therefore, using the L2 norm helps in aligning with these statistical assumptions. Thanks for your insightful comment and careful inspection of the manuscript, and we are sorry for the typo error as regards the superscripts and subscripts. We choose the L2 norm because it is a common measure of the "closeness" between two vectors. In the context of our proposed denoising algorithm, the L2 norm is used to measure the difference between the original image and the denoised image. Also, in our denoising method, we assume that the noise follows a Gaussian distribution, and therefore, using the L2 norm helps in aligning with these statistical assumptions
Point 4: Different wavelet basis functions are suitable for different types of signals and noise. Why is the wavelet basis described in Formula 7 chosen?
Response 4: Thanks so much for the effort imputed to our work to ensure that it comes out in its best form. We have added the description of the wavelet basis in line 327 to 331. The wavelet basis described in (7) was chosen because it is the coefficients of the resulting signal decomposed and this was used to reconstruct the image signal. Due to their ability to represent signal features at different scales, they are effective for noise removal. The localized nature of wavelets allows for the separation of signal and noise components, enabling the removal of unwanted noise while preserving important signal features.
Point 5: You only described the computational complexity of the proposed algorithm. To emphasize the advantages of the proposed algorithm in terms of computational time, please compare the computational time with other algorithms in a table format.
Response 5: We very much appreciate this helpful comment and agree that comparison of the computational time with other algorithms is of potential importance for the presented analysis. We have included the table showing the comparison time with other algorithms in Section 4.2.5 of the revised manuscript.
Point 6: Recently, the diffusion denoising generative model has achieved good performance in Magnetic resonance imaging (MRI). However, the relevant content was not discussed in the manuscript. Please discuss literatures related to diffusion denoising generative model and add them to Table 2 and Table 3.
Response 6: Thank you for this suggestion. It would have been interesting to explore this aspect. However, in the case of our study, it seems slightly out of scope because the diffusion-denoising generative model experimented with according to Zhanxiong Wu et al 2023, introduces a self-attention mechanism with thousands of images for training, a kind of supervised learning. In line 90 and 91, we stated clearly that our algorithm falls under the category of filtering and regularization techniques using neighborhood pixels and not diffusion-denoising generative models. We have also included the diffusion denoising generative model in our literature review section from lines 220 to 230. Our algorithm deals with denoising brain MRI, and therefore cannot be added to the tables as suggested.
Point 7: Tables 2 and 3 only show that the results of Slice 20 are weakly persuasive. More samples and even slices of the entire dataset should be denoised and presented with the average and standard deviation of the metrics
Response 7: We are grateful to the reviewer for your time spent on this paper. It is true that we only made use of slice 20 from the Brainweb dataset. The reason
is that slices used for comparison differ from article to article. And we can only put together the journal that used the same slices together. However, we have made further analysis on another slice to show the superiority of our proposed algorithm to other existing denoising models, and a new table tagged Table 4, has been created and added to the manuscript because it cannot be merged with Table 2 as suggested.
Point 8: It is obviously not enough to select only three subjects from the OASIS Cross-sectional data set to validate the effectiveness of the proposed algorithm. Please evaluate more cases.
Response 8: Thanks for your observation.: We have evaluated altogether 7 subjects for the OASIS Cross-sectional datasets and their performance based on the four metrics (SSIM, PNSR, RMSE, FSIM) were analyzed and plotted. This has been added to section 4 of our manuscript.
Point 9: With the development of MRI technology, the emergence of multi coil imaging mode makes noise become complex non central Chi-squared distribution noise rather than Rician noise in single coil imaging. Please consider more types of noise in the manuscript.
Response 9: We appreciate your insightful comment. Section 2.1.3 lines 153 and 154 clearly stated that “However, in MRI, as our case study, existing literature [11] has established that the prominent noises present in it are Gaussian and Rician Noise.” As established by Goyal et al; 2018 [11], the prominent noises present in MRI are Gaussian and Rician Noise. They are therefore used as the reference noise added.
Point 10: The denoising methods for magnetic resonance images can be roughly divided into four categories: filtering methods, transform domain methods, statistical modeling methods, and regularization methods. Therefore, please cluster the algorithms mentioned in section 2.3.
Response 10: Thanks for your observations and contributions.Thanks for your observations and contributions.
Point 11: The description of image format conversion appears in the manuscript, which is considered redundant. Your work focuses on denoising, so please reduce the text related to image conversion.
Response 11: Thanks so much for your observation. We have reduced the text into a single paragraph, by removing the redundant section and summarizing it as indicated in lines 44 to line 51 of the revised manuscript.

Reviewer 2 Report
The authors presented a method called Regularized Neighborhood Pixel Similarity Wavelet to denoise MRI images. The study used synthetic noise to test their method. Several qualification metrics were used to test the performance.
1. Why do authors state "denoising of nifti images". This algorithm only works for nifti? Not DICOM or Tiff stacks? I don't understand this.
2. Section 2 is mostly common knowledge in the field now. This section should be deleted and not be in a research article. The authors can summarize this section into one paragraph in the introduction.
3. What is the SSIM, PSNR and other metric of the original image? IT needs to be compared to other values as the ground truth.
4. The authors should use other real MRI images to test the performance.
5. There is no ethical statement regarding human brain scans. Where did the authors obtain the data?
Author Response
Response to Reviewer 2 comments
Point 1: Why do authors state "denoising of nifti images". This algorithm only works for nifti? Not DICOM or Tiff stacks? I don't understand this.
Response 1: Thank you so much for the effort imputed to review our work. There are several formats of storing medical images, and although each file format serves different purposes for various use cases, DICOM is overwhelmingly the most widely employed medical image format. Other formats are frequently converted to DICOM for additional processing. Nevertheless, when it comes to a natively 3D system, NRRD and NIFTI are often the preferred choices as they can directly store multidimensional data. Our goal is to use the best format that will enable us to get all features we need for our model to predict accurately and also for other post-processing methods, and that was what gave birth to the title of the article. Therefore, our proposed algorithm is designed in such a way that any format of images can be used with it, be it Nifti, DICOM, or tiff stacks as stated in lines 266 and 267 “Thus, making it a versatile method that can be applied to images of any size and resolution.
Point 2: Section 2 is mostly common knowledge in the field now. This section should be deleted and not be in a research article. The authors can summarize this section into one paragraph in the introduction.
Response 2: Thanks so much for your observation. We have reduced the text into a single paragraph, as indicated in lines 44 to line 51 of the revised manuscript.
Point 3: What is the SSIM, PSNR, and other metric of the original image? It needs to be compared to other values as the ground truth.
Response 3: We appreciate the reviewer’s helpful comment. We have evaluated altogether 7 subjects for the OASIS Cross-sectional datasets and their performance based on the four metrics (SSIM, PNSR, RMSE, FSIM) were analyzed and plotted. This has been added to section 4 of our manuscript.
Point 4: The authors should use other real MRI images to test the performance.
Response 4: Thanks so much for your observation. We appreciate your suggestion and we have evaluated our proposed algorithm on a Clinical Leg MRI and this has been added to section 4.2.4 of the revised manuscript.
Point 5: There is no ethical statement regarding human brain scans. Where did the authors obtain the data?
Response 5: Thanks for your time. This has been included under the “Informed Consent Statement” which is shown after the conclusion section of our manuscript.
